# ConML: A Universal Meta-Learning Framework with Task-Level Contrastive Learning

## Abstract

Meta-learning enables learning systems to adapt quickly to new tasks, similar to humans. To emulate this human-like rapid learning and enhance alignment and discrimination abilities, we propose ConML, a universal meta-learning framework that can be applied to various meta-learning algorithms without relying on specific model architectures nor target models. The core of ConML is task-level contrastive learning, which extends contrastive learning from the representation space in unsupervised learning to the model space in meta-learning. By leveraging task identity as an additional supervision signal during meta-training, we contrast the outputs of the meta-learner in the model space, minimizing inner-task distance (between models trained on different subsets of the same task) and maximizing inter-task distance (between models from different tasks). We demonstrate that ConML integrates seamlessly with optimization-based, metric-based, and amortization-based meta-learning algorithms, as well as in-context learning, resulting in performance improvements across diverse few-shot learning tasks. Code is provided at https://anonymous.4open.science/r/conml_ano-3372.

## 1 Introduction

Meta-learning, or "learning to learn" (Schmidhuber, 1987; Thrun & Pratt, 1998), is a powerful paradigm designed to enable learning systems to adapt quickly to new tasks. During the meta-training phase, a meta-learner simulates learning across a variety of relevant tasks to accumulate knowledge on how to adapt effectively. In the meta-testing phase, this learned adaptation strategy is applied to previously unseen tasks. The adaptation is typically accomplished by the meta-learner, which, given a set of task-specific training examples, generates a predictive model tailored to that task. Meta-learning has been widely applied to important learning problems such as few-shot learning (Finn et al., 2017; Wang et al., 2020), reinforcement learning (Yu et al., 2020; Nagabandi et al., 2019), and neural architecture search (Elsken et al., 2020; Shaw et al., 2019).

Despite the success of meta-learning, there remains room for improvement in how models generalize to new tasks. Human learning leverages two key cognitive abilities—alignment and discrimination—which are essential for rapid adaptation (Hummel, 2013; Chen, 2012; Christie, 2021). Alignment refers to integrating different perspectives of an object to form a coherent understanding (Christian, 2021), while discrimination involves distinguishing between similar stimuli to respond appropriately only to relevant inputs (Robbins, 1970). Incorporating these abilities into meta-learning models could enhance their adaptability and precision.

Several existing approaches attempt to leverage alignment and discrimination in meta-learning. Some methods assume that task-specific target models for meta-training tasks are available, allowing meta-training to be supervised by aligning the learned model with the target model, either through model weight alignment (Wang & Hebert, 2016; Wang et al., 2017) or knowledge distillation (Ye et al., 2022). However, these methods have limited applicability, as learning target models can be computationally expensive and, in many real-world problems, such target models are not readily available. Fei et al. (2021) and Wang et al. (2023) introduce alignment into meta-learners by aligning classifiers in few-shot classification. However, their reliance on a static pool of base classes for meta-training limits their flexibility in more dynamic or diverse tasks. Similarly, Gondal et al. (2021) and Mathieu et al. (2021) explore contrastive representations for neural processes, but their methods are tied to specific models, reducing their generalizability.

In this paper, we propose ConML, a universal meta-learning framework that employs task-level contrastive learning to enhance both alignment and discrimination abilities. Our approach is applicable to various meta-learning algorithms without relying on specific model architectures, target models, or extensive modifications. By treating tasks similarly to how contrastive learning handles unlabeled samples, ConML contrasts the outputs of the meta-learner based on task identity. Positive pairs consist of different subsets of the same task, while negative pairs come from different tasks, with the objective of minimizing inner-task distance (alignment) and maximizing inter-task distance (discrimination). ConML is efficient, requiring no additional data or retraining, and is learner-agnostic that can be integrated into diverse representative meta-learning algorithms from different categories including optimization-based (e.g., MAML (Finn et al., 2017)), metric-based (e.g., ProtoNet (Snell et al., 2017)), and amortization-based (e.g., Simple CNAPS (Bateni et al., 2020)). We also demonstrate how ConML enhances in-context learning (Brown et al., 2020) within the meta-learning paradigm.

Our contributions can be summarized as below:

- We propose a universal meta-learning framework, ConML, that emulates cognitive alignment and discrimination abilities in meta-learning, helping to bridge the gap between the fast learning capabilities of humans and meta-learners.

- We extend contrastive learning from the representation space in unsupervised learning to the model space in meta-learning. By introducing model representations for various types of meta-learners, ConML integrates efficiently into episodic training.

- We empirically show that ConML universally improves a wide range of meta-learning algorithms with minimal implementation cost on diverse few-shot learning problems and in-context learning.

In the following sections, we first review related work and introduce the general learning to learn process in Section 2, setting the foundation for how we incorporate alignment and discrimination into meta-learning. In Section 3, we first present the general framework of learnig with ConML and then provide the specifications of different types of meta-learning approaches. Section 4 presents experimental results, demonstrating the effectiveness of ConML.

## 2 META-LEARNING: RELATED WORKS AND PRELIMINARIES

Meta-learning, or "learning to learn," focuses on improving the learning algorithm itself (Schmidhuber, 1987). Popular meta-learning approaches can be broadly categorized into three types (Bronskill et al., 2021): (i) Optimization-based approaches (Andrychowicz et al., 2016; Finn et al., 2017; Nichol et al., 2018), which focus on learning better optimization strategies for adapting to new tasks; (ii) Metric-based approaches (Vinyals et al., 2016; Snell et al., 2017; Sung et al., 2018), which leverage learned similarity metrics; and (iii) Amortization-based approaches (Garnelo et al., 2018; Requeima et al., 2019; Bateni et al., 2020), which aim to learn a shared representation across tasks, amortizing the adaptation process by using neural networks to directly infer task-specific parameters from the training set.

All of these meta-learning approaches utilize episodic training, where the meta-learner is trained across multiple tasks (episodes). Recently, several works have focused on optimizing task sampling strategies by adjusting the task schedule based on task difficulty and diversity (Agarwal & Singh, 2023; Han et al., 2021; Zhang et al., 2022; Liu et al., 2020; Kumar et al., 2023). In contrast, ConML does not alter the task sampling process. Instead, it incorporates task-level contrastive learning within the episodic training framework to enhance the meta-learner's alignment and discrimination abilities. Moreover, ConML can be used alongside task sampling strategies, further increasing its versatility.

Formally, let $\mathcal{L}(\mathcal{D}; h)$ represent the loss when evaluating a model $h$ on a dataset $\mathcal{D}$ using a loss function $\ell(y, \hat{y})$ (e.g., cross-entropy or mean squared error). Let $g(; \theta)$ be a meta-learner that maps a dataset $\mathcal{D}$ to a model $h$, i.e., $h = g(\mathcal{D}; \theta)$. Given a distribution of tasks $p(\tau)$, where each task $\tau$ consists of a training set $\mathcal{D}_\tau^{\text{tr}} = (x_{\tau,i}, y_{\tau,i})_{i=1}^n$ and a validation set $\mathcal{D}_\tau^{\text{val}} = (x_{\tau,i}, y_{\tau,i})_{i=n+1}^m$, the objective of meta-learning is to train $g(; \theta)$ to generalize well to a new task $\tau'$ sampled from $p(\tau')$. The model's performance on the new task is evaluated by $\mathcal{L}(\mathcal{D}_{\tau'}^{\text{val}}; g(\mathcal{D}_{\tau'}^{\text{tr}}; \theta))$.

In meta-training, the meta-learner $g(; \theta)$ is optimized through a series of episodes, each consisting of a batch $\boldsymbol{b}$ of $B$ tasks. The goal is to minimize an episodic loss $L_v$. A common objective is to minimize the validation loss, given by $\mathbb{E}_{\tau \sim p(\tau)} \mathcal{L}(\mathcal{D}_\tau^{\text{val}}; g(\mathcal{D}_\tau^{\text{tr}}; \theta))$. As outlined in Algorithm 1, in each episode, $B$ tasks are sampled from $p(\tau)$ to form the batch $\boldsymbol{b}$, and the validation loss for each task is aggregated as the supervision signal: $L_v = \frac{1}{B} \sum_{\tau \in \boldsymbol{b}} \mathcal{L}(\mathcal{D}_\tau^{\text{val}}; g(\mathcal{D}_\tau^{\text{tr}}; \theta))$, which is used to update $\theta$. While different meta-learning algorithms may implement their own specific functions within $g$ and $L_v$, they all share this same episodic training framework to develop the ability to generalize across tasks.

---

**Algorithm 1** Meta-Training.

---

**while** Not converged **do**
    Sample a batch of tasks $\boldsymbol{b} \sim p^B(\tau)$.
    **for** All $\tau \in \boldsymbol{b}$ **do**
        Get task-specific model $h_\tau = g(\mathcal{D}_\tau^{\text{tr}}; \theta)$;
        Get validation loss $\mathcal{L}(\mathcal{D}_\tau^{\text{val}}; h_\tau)$;
    **end for**
    $L_v = \frac{1}{B} \sum_{\tau \in \boldsymbol{b}} \mathcal{L}(\mathcal{D}_\tau^{\text{val}}; g(\mathcal{D}_\tau^{\text{tr}}; \theta))$
    Update $\theta$ by $\theta \leftarrow \theta - \nabla_\theta L_v$.
**end while**

---

## 3 META-LEARNING WITH CONML

Now, we introduce our ConML (Figure 1 which equip meta-learners with the desired alignment and discrimination ability via task-level CL. We first present the general framework of ConML based on episodic training in Section 3.1, followed by specifications for the three main streams of meta-learning approaches in Section 3.2.

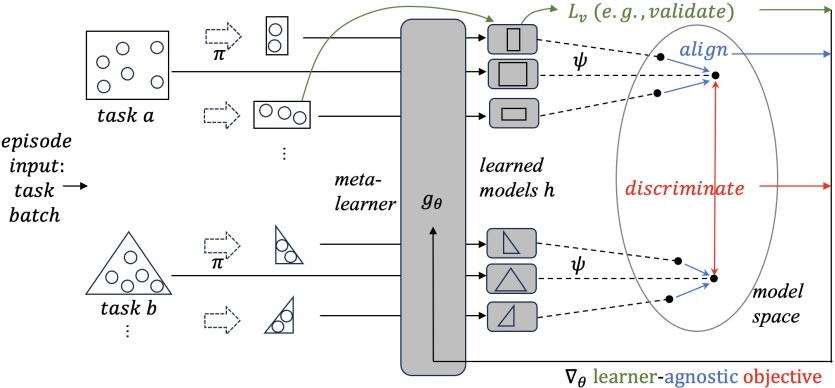

Figure 1: Illustration of ConML. In each episode, given a batch of tasks sampled from the meta-training dataset, ConML first samples subsets from each task using the sub-sampling strategy $\pi$. These subsets are independently fed into the meta-learner $g_\theta$, and the resulting models are projected into the model space using a learner-specific projection $\psi$ to perform alignment and discrimination. Combined with the original episodic loss $L_v$ (e.g., validation loss), this forms a learner-agnostic objective for optimization.

### 3.1 A GENERAL FRAMEWORK

To enhance the alignment and discrimination abilities of meta-learning, we draw inspiration from Contrastive Learning (CL) (Oord et al., 2018; Chen et al., 2020; Wang & Isola, 2020). CL focuses on learning representations that are invariant to irrelevant details while preserving essential information. This is achieved by maximizing alignment and discrimination (uniformity) in the representation space (Wang & Isola, 2020). While most existing studies focus on sample-wise contrastive learning in the representation space via unsupervised learning (Wu et al., 2018; Hjelm et al., 2018; Bachman et al., 2019; Tian et al., 2020; He et al., 2020; Chen et al., 2020; Khosla et al., 2020), we extend CL to the model space in meta-learning. Specifically, we design a taks-level CL in the model space, where alignment is achieved by minimizing the inner-task distance (i.e., the distance between models trained on different subsets of the same task), and discrimination is achieved by maximizing the inter-task distance (i.e., the distance between models from different tasks). The detailed procedures of ConML are introduced below.

**Obtaining Model Representation.** To train the meta-learner $g$, the distances $d^{\text{in}}$, $d^{\text{out}}$ are measured in the output space of $g$, also referred to as the model space $\mathcal{H}$. A practical approach is to represent the model $h = g(\mathcal{D}; \theta) \in \mathcal{H}$ as a fixed-length vector $\boldsymbol{e} \in \mathbb{R}^d$, and then compute the distances using an explicit distance function $\phi(\cdot, \cdot)$ (e.g., cosine distance). Note that $\mathcal{H}$ is learner-specific, so to generalize across different learners to form a learner-agnostic framework, we introduce a projection function $\psi : \mathcal{H} \to \mathbb{R}^d$ to obtain the model representations $\boldsymbol{e} = \psi(h)$, and then the proceedings are learner-agnostic. The details of $\mathcal{H}$ and $\psi$ will be explained and specified for different meta-learners in Section 3.2.

**Obtaining Inner-Task Distance.** During meta-training, the combined dataset $\mathcal{D}_\tau^{\text{tr}} \cup \mathcal{D}_\tau^{\text{val}}$ contains all the available information about task $\tau$. The meta-learner is expected to produce similar models when trained on any subset $\kappa$ of this dataset. Moreover, models trained on subsets should resemble the model learned from the full dataset $\mathcal{D}_\tau^{\text{tr}} \cup \mathcal{D}_\tau^{\text{val}}$. For $\forall \kappa \subseteq \mathcal{D}_\tau^{\text{tr}} \cup \mathcal{D}_\tau^{\text{val}}$, we expect $\boldsymbol{e}_\tau^\kappa = \boldsymbol{e}_\tau^*$, where $\boldsymbol{e}_\tau^\kappa = \psi(g(\kappa; \theta)), \boldsymbol{e}_\tau^* = \psi(g(\mathcal{D}_\tau^{\text{tr}} \cup \mathcal{D}_\tau^{\text{val}}; \theta))$. The inner-task distance $d_\tau^{\text{in}}$ for each task $\tau$ is computed as:

$$d_\tau^{\text{in}} = (1/K) \cdot \sum\nolimits_{k=1}^{K} \phi(\boldsymbol{e}_\tau^{\kappa_k}, \boldsymbol{e}_\tau^*), \ s.t., \kappa_k \sim \pi_\kappa(\mathcal{D}_\tau^{\text{tr}} \cup \mathcal{D}_\tau^{\text{val}}), \tag{1}$$

where $\{\kappa_k\}_{k=1}^K$ are $K$ subsets sampled from $\mathcal{D}_\tau^{\text{tr}} \cup \mathcal{D}_\tau^{\text{val}}$ using a specific sampling strategy $\pi_\kappa$. In each episode, given a batch $\mathbf{b}$ of task containing $B$ tasks, the overall inner-task distance is averaged as $d^{\text{in}} = \frac{1}{B} \sum_{\tau \in \boldsymbol{b}} d_\tau^{\text{in}}$.

**Obtaining Inter-Task Distance.** Since the goal of meta-learning is to improve performance on unseen tasks, it is crucial for the meta-learner $g$ to generalize well across diverse tasks. Given the natural assumption that different tasks require distinct task-specific models, it is essential that $g$ can learn to differentiate between tasks—i.e., possess strong discrimination capabilities. To enhance task-level generalization, we define the inter-task distance $d^{\text{out}}$, which should be maximized to encourage $g$ to learn distinct models for different tasks. Specifically, for any two tasks $\tau \neq \tau'$ during meta-training, we aim to maximize the distance between their respective representations, $\boldsymbol{e}_\tau^*$ and $\boldsymbol{e}_{\tau'}^*$. To make this practical within the mini-batch episodic training paradigm, we compute $d^{\text{out}}$ across a batch of tasks sampled in each episode:

$$d^{\text{out}} = (1/B(B-1)) \cdot \sum\nolimits_{\tau \in \boldsymbol{b}} \sum\nolimits_{\tau' \in \boldsymbol{b} \setminus \tau} \phi(\boldsymbol{e}_\tau^*, \boldsymbol{e}_{\tau'}^*). \tag{2}$$

**Training Procedure.** We optimize ConML w.r.t. the combination of the original episodic loss $L_v$ and contrastive meta-objective $d^{\text{in}} - d^{\text{out}}$:

$$L = L_v + \lambda(d^{\text{in}} - d^{\text{out}}). \tag{3}$$

The training procedure of ConML is provided in Algorithm 2. Compared to Algorithm 1, ConML introduces additional computations for $\psi(g(\mathcal{D}; \theta))$ a total of $K + 1$ times per episode. However, $\psi$ is implemented as a lightweight function (e.g., extracting model weights), and $g(\mathcal{D}; \theta)$ is already part of the standard episodic training process, with multiple evaluations of $g(\mathcal{D}; \theta)$ being parallelizable. As a result, ConML incurs only a minimal increase in computational cost.

---

**Algorithm 2** Meta-Training with ConML.

---

**while** Not converged **do**
    Sample a batch of tasks $\boldsymbol{b} \sim p^B(\tau)$.
    **for** All $\tau \in \boldsymbol{b}$ **do**
        †Sample $\kappa_k$ from $\pi_\kappa(\mathcal{D}_\tau^{\text{tr}} \cup \mathcal{D}_\tau^{\text{val}})$ for $k \in \{1, 2, \cdots, K\}$;
        †Get model representation $\boldsymbol{e}_\tau^{\kappa_k} = \psi(g(\kappa_k; \theta))$;
        †Get model representation $\boldsymbol{e}_\tau^* = \psi(g(\mathcal{D}_\tau^{\text{tr}} \cup \mathcal{D}_\tau^{\text{val}}; \theta))$;
        †Get inner-task distance $d_\tau^{\text{in}}$ by equation 1;
        Get task-specific model $h_\tau = g(\mathcal{D}_\tau^{\text{tr}}; \theta)$;
        Get episodic loss $L_v$;
    **end for**
    †Get $d^{\text{in}} = \frac{1}{B} \sum_{\tau \in \boldsymbol{b}} d_\tau^{\text{in}}$ and $d^{\text{out}}$ by equation 2;
    Get loss $L$ by equation 3;
    Update $\theta$ by $\theta \leftarrow \theta - \nabla_\theta L$.
**end while**

---

"†" indicates additional steps introduced by ConML to integrate into standard episodic training procedure of meta-learning.

## 3.2 Integrating ConML with Mainstream Meta-Learning Approaches

ConML is universally applicable to enhance any meta-learning algorithm that follows episodic training. It does not depend on a specific form of $g$ or $L_v$ and can be used alongside other forms of task-level information. Next, we provide the specifications of $\mathcal{H}$ and $\psi(g(\mathcal{D}, \theta))$ to obtain model representations for implementing ConML. We illustrate examples across different categories of meta-learning algorithms, including optimization-based, metric-based, and amortization-based approaches.

These examples are explicitly represented by model weights, as summarized in Table 1. Appendix A provides the detailed procedures for integrating ConML with various meta-learning algorithms. We also demonstrate how ConML enhances in-context learning within the meta-learning paradigm in Appendix B.

Table 1: Specifications of integrating ConML with mainstream meta-learning approaches.

| Category | Examples | Meta-learner $g(\mathcal{D};\theta)$ | Model representation $\psi(g(\mathcal{D};\theta))$ |
|---|---|---|---|
| Optimization -based | MAML, Reptile | Update model weights $\theta - \nabla_\theta \mathcal{L}(\mathcal{D};h_\theta)$ | $\theta - \nabla_\theta \mathcal{L}(\mathcal{D};h_\theta)$ |
| Metric -based | ProtoNet, MatchNet | Build classifier with $\{(\{f_\theta(x_i)\}_{x_i \in \mathcal{D}_j}, \text{label } j)\}_{j=1}^N$ | Concatenate $[\frac{1}{|\mathcal{D}_j|}\sum_{x_i \in \mathcal{D}_j} f_\theta(x_i)]_{j=1}^N$ |
| Amortization -based | CNPs, CNAPs | Map $\mathcal{D}$ to model weights by $H_\theta(\mathcal{D})$ | $H_\theta(\mathcal{D})$ |

**With Optimization-Based Methods.** The representative algorithm of optimization-based meta-learning is MAML, which meta-learns an initialization from where gradient steps are taken to learn task-specific models, i.e., $g(\mathcal{D};\theta) = h_{\theta - \nabla_\theta \mathcal{L}(\mathcal{D};h_\theta)}$. Since MAML directly generates the model weights, we use these weights as model representation. Specifically, the representation of the model learned by $g$ given a dataset $\mathcal{D}$ is: $\psi(g(\mathcal{D};\theta)) = \theta - \nabla_\theta \mathcal{L}(\mathcal{D};h_\theta)$. certain optimization-based meta-learning algorithms, such as FOMAML (Finn et al., 2017) and Reptile (Nichol et al., 2018), use first-order approximations of MAML and do not strictly follow Algorithm 1 to minimize validation loss. Nonetheless, ConML can still be incorporated into these algorithms as long as they adhere to the episodic training framework.

**With Metric-Based Methods.** Metric-based algorithms are well-suited for classification tasks. Given a dataset $\mathcal{D}$ for an $N$-way classification task, these algorithms classify based on the distances between input samples $\{\{f_\theta(x_i)\}_{x_i \in \mathcal{D}_j}\}_{j=1}^N$ and their corresponding labels, where $f_\theta$ is a meta-learned encoder and $\mathcal{D}_j$ represents the set of inputs for class $j$. We represent this metric-based classifier by concatenating the mean embeddings of each class in a label-aware order. For example, ProtoNet (Snell et al., 2017) computes the prototype $c_j$, which is the mean embedding of samples in each class: $c_j = \frac{1}{|\mathcal{D}_j|}\sum_{(x_i,y_i) \in \mathcal{D}_j} f_\theta(x_i)$. The classifier $h_{\theta,\mathcal{D}}$ then makes predictions as $p(y = j \mid x) = \exp(-d(f_\theta(x), c_j))/\sum_{j'} \exp(-d(f_\theta(x), c_{j'}))$. Since the outcome model $h_{\theta,\mathcal{D}}$ depends on $\mathcal{D}$ through $\{c_j\}_{j=1}^N$ and their corresponding labels, the representation is specified as $\psi(g(\mathcal{D};\theta)) = [c_1|c_2|\cdots|c_N]$, where $[\cdot|\cdot]$ denotes concatenation.

**With Amortization-Based Methods.** Amortization-based approaches meta-learns a hypernetwork $H_\theta$ that aggregates information from $\mathcal{D}$ to task-specific parameter $\alpha$, which serves as the weights for the main-network $h$, resulting in a task-specific model $h_\alpha$. For example, Simple CNAPS (Bateni et al., 2020) uses a hypernetwork to generate a small set of task-specific parameters that perform feature-wise linear modulation (FiLM) on the convolution channels of the main-network. In ConML, we represent the task-specific model $h_\alpha$ using the task-specific parameters $\alpha$, i.e., the output of the hypernetwork $H_\theta$: $\psi(g(\mathcal{D};\theta)) = H_\theta(\mathcal{D})$.

## 4 EXPERIMENTS

### 4.1 FEW-SHOT REGRESSION

We begin by conducting experiments on synthetic data in a controlled setting to address two key questions: (i) Does training with ConML enable meta-learners to develop alignment and discrimination abilities that generalize to meta-testing tasks? (ii) How do alignment and discrimination individually contribute to meta-learning performance?

We take MAML w/ ConML as example and investigate above questions with few-shot regression problem following the same settings in (Finn et al., 2017). Each task involves regressing from the input to the output of a sine wave, where the amplitude and phase of the sinusoid are varied between tasks. The amplitude varies within $[0.1, 5.0]$ and the phase varies within $[0, \pi]$. This synthetic regression dataset allows us to sample data and adjust the distribution as necessary for analysis.

The implementation of ConML follows a simple intuitive setting: inner-task sampling $K = 1$ and $\pi_\kappa(\mathcal{D}_\tau^{\text{tr}} \cup \mathcal{D}_\tau^{\text{val}}) = \mathcal{D}_\tau^{\text{tr}}$, $\phi(a, b) = 1 - {}^{a \cdot b}/_{\|a\|\|b\|}$ (cosine distance) and $\lambda = 0.1$.

Table 2: Meta-testing and clustering performance on few-shot regression problem.

| Method | MSE (5-shot) | MSE (10-shot) | Silhouette | DBI | CHI |
|---|---|---|---|---|---|
| MAML | $.6771 \pm .0377$ | $.0678 \pm .0022$ | $.1068 \pm .0596$ | $.0678 \pm .0021$ | $31.55 \pm 2.52$ |
| MAML w/ ConML | $\mathbf{.3935} \pm .0100$ | $\mathbf{.0397} \pm .0009$ | $\mathbf{.1945} \pm .0621$ | $\mathbf{.0397} \pm .0009$ | $\mathbf{39.22} \pm 2.61$ |

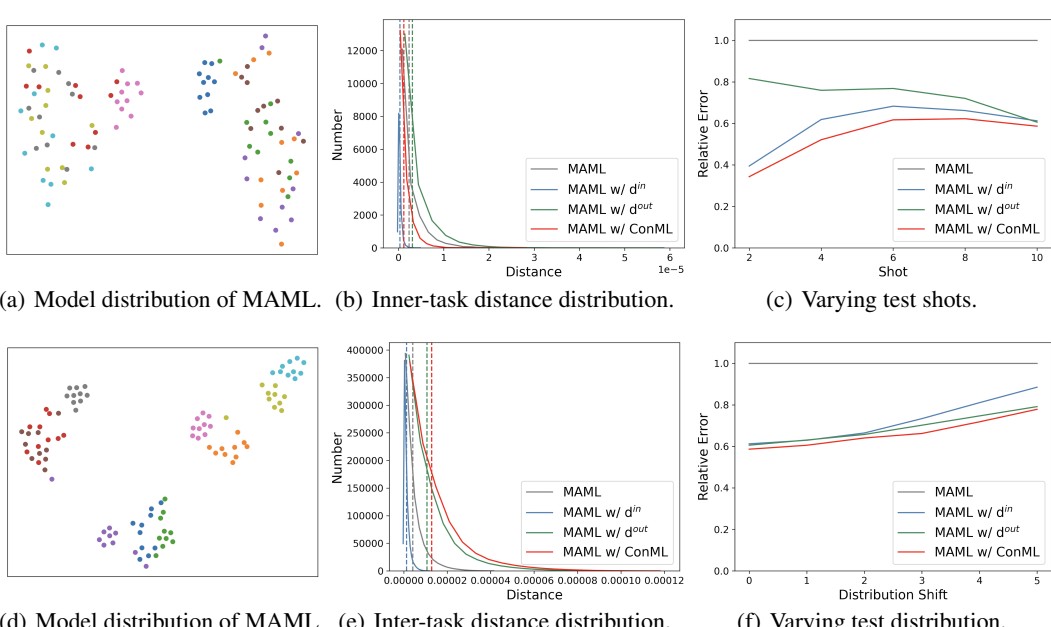

(a) Model distribution of MAML.  (b) Inner-task distance distribution.  (c) Varying test shots.

(d) Model distribution of MAML w/ ConML.  (e) Inter-task distance distribution.  (f) Varying test distribution.

Figure 2: Evaluation of ConML on few-shot regression problem.

**ConML Brings Generalizable Alignment and Discrimination.** If optimizing $d^{\text{in}}$ and $d^{\text{out}}$ does equip meta-learner with generalizable alignment and discrimination, MAML w/ ConML can generate more similar models from different subsets of the same task, while generating more separable models from different tasks. This can be verified by evaluating the clustering performance for model representations $e$. During meta-testing, we randomly sample 10 different tasks. For each task, we sample 10 different subsets, each containing $N = 10$ samples. Using these 100 different training sets $\mathcal{D}^{\text{tr}}$ as input, the meta-learner generates 100 models. Figure 2(a) and 2(d) visualize the distribution of these models, where each point corresponds to the result of a subset and the same color indicates sampled from the same task. It can be obviously observed MAML w/ ConML performs better alignment and discrimination than MAML. To quantity the results, we also evaluate the supervised clustering performance, where task identity is used as label. Table 2 shows the supervised clustering performance of different metrics: Silhouette score (Rousseeuw, 1987), Davies-Bouldin index (DBI) (Davies & Bouldin, 1979) and Calinski-Harabasz index (CHI) (Caliński & Harabasz, 1974). The results indicate that MAML with ConML significantly outperforms standard MAML across all metrics. These findings confirm that training with ConML enables meta-learners to develop alignment and discrimination abilities that generalize to meta-testing tasks.

**Alignment Enhance Fast-Adaptation and Discrimination Enhance Task-Level Generalizability.** We aim to understand the individual contributions of optimizing $d^{\text{in}}$ (alignment) and $d^{\text{out}}$ (discrimination) to meta-learning performance. In conventional unsupervised contrastive learning, both positive and negative pairs are necessary to avoid learning representations without useful information. However, in ConML, the episodic loss $L_v$ plays a fundamental role in "learning to learn," while the

contrastive objective serves as additional supervision to enhance alignment and discrimination. Thus, we consider two variants of ConML: MAML w/ $d^{\text{in}}$ which optimize $L_v$ and $d^{\text{in}}$, MAML w/ $d^{\text{out}}$ which optimize $L_v$ and $d^{\text{out}}$. During meta-testing, we randomly sample 1000 different tasks, with 10 different subsets (each containing $N = 10$ samples) per task. These subsets are aggregated into a single set of $N = 100$ to obtain $e_\tau^*$ for each task. Figure 2(b) and 2(e) visualize the distribution of $d^{\text{in}}$ and $d^{\text{out}}$ respectively, where the dashed lines mark mean values. Smaller $d^{\text{in}}$ means better alignment and larger $d^{\text{out}}$ means better discrimination. We can find that the alignment and discrimination abilities are separable, generalizable, and that ConML effectively couples both. Figure 2(c) shows the testing performance given different numbers of examples per task (shot), while the meta-leaner is trained with a fixed $N = 10$. The results indicate that the improvement from alignment (MAML w/ $d^{\text{in}}$) is more pronounced in few-shot scenarios, highlighting its close relationship with fast-adaptation. Figure 2(f) shows the out-of-distribution testing performance. Meta-trained on tasks with amplitudes uniformly distributed over $[0.1, 5]$, meta-testing is performed on tasks with amplitudes uniformly distributed over $[0.1 + \delta, 5 + \delta]$, where $\delta$ is shown on the $x$-axis. As the distribution gap increases, the improvement from discrimination (MAML w/ $d^{\text{out}}$) is more significant than from alignment (MAML w/ $d^{\text{in}}$), indicating that discrimination plays a critical role in task-level generalization. ConML leverages the benefits of both alignment and discrimination.

## 4.2 Few-Shot Image Classification

Here, we evaluate the meta-learning performance on few-shot image classification problem follow existing works (Vinyals et al., 2016; Finn et al., 2017; Bateni et al., 2020). We use two few-shot image classification benchmarks: miniImageNet (Vinyals et al., 2016) and tieredImageNet (Ren et al., 2018), evaluating on 5-way 1-shot and 5-way 5-shot tasks. We also evaluate on a large-scale dataset, META-DATASET (Triantafillou et al., 2020), whose results are provided in Appendix C.

We consider representative meta-learning algorithms from different categories, including optimization-based: **MAML** (Finn et al., 2017), **FOMAML** (Finn et al., 2017), **Reptile** (Nichol et al., 2018); metric-based: **MatchNet** (Vinyals et al., 2016), **ProtoNet** (Snell et al., 2017); and amortization-based: **SCNAPs** (Simple CNAPS) (Bateni et al., 2020). We evaluate the meta-learning performance of each algorithm in its original form (w/o ConML) and after incorporating ConML into the training process (w/ ConML). The implementation of ConML follows the general procedure described in Algorithm 2 and the specification for corresponding category in Section 3.2. In addition, we assess the impact of ConML on a state-of-the-art few-shot image classification method utilizing in-context learning, **CAML** (Fifty et al., 2024). The implementation details for equipping in-context learners with ConML are provided in Appendix B.

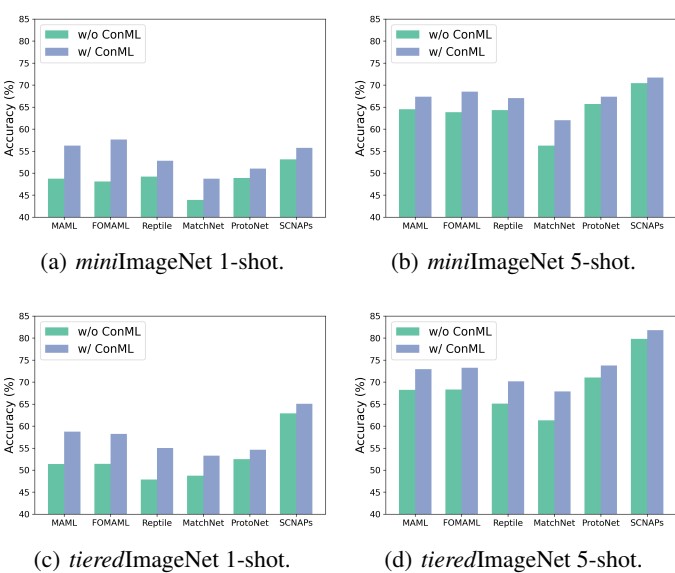

(a) *mini*ImageNet 1-shot.

(b) *mini*ImageNet 5-shot.

(c) *tiered*ImageNet 1-shot.

(d) *tiered*ImageNet 5-shot.

Figure 3: Few-shot image classification performance.

Since we focus on comparing the improvement of each algorithm with and without ConML, rather than comparing different algorithms directly, we follow the originally reported settings for each baseline. For ConML, all baselines share the same hyperparameter settings: task batch size $B = 32$, inner-task sampling $K = 1$, and $\pi_\kappa(\mathcal{D}_\tau^{\text{tr}} \cup \mathcal{D}_\tau^{\text{val}}) = \mathcal{D}_\tau^{\text{tr}}$. We use cosine distance for $\phi(a, b) = 1 - {a \cdot b}/{\|a\|\|b\|}$ and set $\lambda = 0.1$. Other hyperparameters related to model architecture and training procedure remain consistent with their original settings. Note that setting $K = 1$ and $\pi_\kappa(\mathcal{D}_\tau^{\text{tr}} \cup \mathcal{D}_\tau^{\text{val}}) = \mathcal{D}_\tau^{\text{tr}}$ represents the simplest and most efficient implementation, referred to as ConML ($K = 1$) in Appendix A. In this setup, the only additional computation compared to Algorithm 1 is $g(\mathcal{D}_\tau^{\text{tr}} \cup \mathcal{D}_\tau^{\text{val}}; \theta)$, resulting in similar time consumption, as shown in Section 4.4.1.

Figure 3 shows the results on *mini*ImageNet and *tiered*ImageNet respectively (numerical results are provided in Table 9 and 10 in Appendix C). The relative gain is calculated as the sum of the 1-shot and 5-shot accuracy improvements. The significant relative gain, combined with comparable time consumption (Appendix C), demonstrates that ConML offers universal improvements across different meta-learning algorithms with minimal additional overhead.

Table 3: Testing accuracy (%) of CAML and CAML w/ ConML on few-shot image classification.

| Benchmark | *mini*ImageNet | | *tiered*ImageNet | | Aircraft | | CIFAR-fs | |
|---|---|---|---|---|---|---|---|---|
| Shots | n=1 | n=5 | n=1 | n=5 | n=1 | n=5 | n=1 | n=5 |
| CAML | 96.2 | 98.6 | 95.4 | **98.1** | 63.3 | 79.1 | 70.8 | 85.5 |
| CAML w/ ConML | **97.0** | **98.9** | **96.6** | **98.2** | **65.8** | **81.5** | **72.3** | **86.1** |

Futher, we equip the state-of-the-art meta-learning method CAML (Fifty et al., 2024) with our ConML as **CAML w/ ConML**. CAML is based on in-context learning and consists of a feature extractor and an in-context learner (transformer). The feature extractor is pre-trained and remains frozen during meta-training, meaning ConML does not affect it. Instead, ConML influences the training of the in-context learner, following the procedure described in Appendix B. We adopt the same experimental settings as Fifty et al. (2024), where the feature extractor is a ViT-based model with pre-trained parameters, and the in-context learner is trained on ImageNet-1k, Fungi, MSCOCO, and WikiArt. In Fifty et al. (2024), this process is referred to as "large-scale pretraining," but in our case, it is treated as the meta-training process for ConML. We also introduce ConML with the efficient settings of $K = 1$ and $\pi = $ random half. All other training and evaluation settings remain consistent with the public code provided by Fifty et al. (2024). The results, presented in Table 3, show 5-way $n$-shot performance. We find that CAML with ConML consistently outperforms CAML without it, further demonstrating that ConML is a learner-agnostic approach that can enhance even state-of-the-art methods.

### 4.3 FEW-SHOT MOLECULAR PROPERTY PREDICTION

Few-shot molecular property prediction is another important testbed for meta-learning methods (Altae-Tran et al., 2017; Guo et al., 2021; Wang et al., 2021; Chen et al., 2022; Schimunek et al., 2023). We use FS-Mol (Stanley et al., 2021), a widely studied benchmark consisting of a large number of diverse tasks. We follow the public data split provided in Stanley et al. (2021). Each training set contains 64 labeled molecules and can be imbalanced, where the number of labeled active and inactive molecules may not be equal. The remaining molecules in each task form the validation set.

We consider the following meta-learning approaches: PAR (Wang et al., 2021), ADKF-IFT (Chen et al., 2022), and PACIA (Wu et al., 2024). Note that MHNfs (Schimunek et al., 2023) is excluded from the comparison, as it uses additional reference molecules from external datasets, which would result in an unfair comparison. All baselines share the same encoder provided by the benchmark, which maps molecular graphs to embedding vectors and is meta-trained from scratch. The performance is evaluated by $\Delta$AUPRC (change in area under the precision-recall curve) w.r.t. a random classifier (Stanley et al., 2021), averaged across meta-testing tasks.

We incoporate ConML into state-of-the-art method PACIA, an amortization-based meta-learner, as **PACIA w/ ConML**. The embedding vectors from $\mathcal{D}^{\text{tr}}$ are input into the hypernetwork, and the output modulates the embedding vectors through FiLM. We set the hyperparameters for ConML as follows: $B = 16$, $\phi(a, b) = 1 - {a \cdot b}/{\|a\|\|b\|}$ (cosine distance), and $\lambda = 0.1$. For the sampling

strategy $\pi_\kappa$ and the number of times $K$, we sample subsets of different sizes for each task, specifically $m \in 4, 8, 16, 32, 64$, with $128/m$ iterations for each size. Each subset contains $m/2$ positive and $m/2$ negative samples, selected randomly. All other hyperparameters related to model structure and training follow the default settings from the benchmark (Stanley et al., 2021).

Table 4 presents the results. PACIA w/ ConML outperforms the state-of-the-art approach across all meta-testing scenarios with different shots. Comparing PACIA w/ ConML to the original PACIA, the impact of ConML is notably significant.

Table 4: Few-shot molecular property prediction performance ($\Delta$AUPRC) on FS-Mol.

|  | 2-shot | 4-shot | 8-shot | 16-shot |
|---|---|---|---|---|
| MAML | $.009 \pm .006$ | $.125 \pm .009$ | $.146 \pm .007$ | $.159 \pm .009$ |
| PAR | $.124 \pm .007$ | $.140 \pm .005$ | $.149 \pm .009$ | $.164 \pm .008$ |
| ProtoNet | $.117 \pm .006$ | $.142 \pm .007$ | $.175 \pm .006$ | $.206 \pm .008$ |
| CNP | $.139 \pm .004$ | $.155 \pm .008$ | $.174 \pm .006$ | $.187 \pm .009$ |
| ADKF-IFT | $.131 \pm .007$ | $.166 \pm .005$ | $.202 \pm .006$ | $.234 \pm .009$ |
| PACIA | $.142 \pm .007$ | $.169 \pm .006$ | $.205 \pm .008$ | $.236 \pm .008$ |
| PACIA w/ ConML | $\mathbf{.175} \pm .006$ | $\mathbf{.196} \pm .006$ | $\mathbf{.218} \pm .005$ | $\mathbf{.241} \pm .007$ |

### 4.4 MODEL ANALYSIS OF CONML

As mentioned in Section 4.2, ConML uses a simple and intuitive configuration across all baselines in the experiments, demonstrating significant improvement even with minimal hyperparameter tuning. Fully unlocking the potential of ConML through hyperparameter optimization presents an interesting but challenging task due to the high dimensionality of its settings. These settings include surrogate forms of contrastive loss, the distance function $\phi$, the weight parameter $\lambda$, subset sampling strategy $\pi_\kappa$, the number of subsets $K$, task batch size $B$, and batch sampling strategy, all of which may vary for each specific meta-learner.

In this section, we explore the impact of key ConML settings: (1) the number of subset samples $K$, which influences the model's complexity, and (2) the contrastive loss, including the distance function $\phi$, the weighting factor $\lambda$, and the use of InfoNCE as a replacement for $(d^{\text{in}} - d^{\text{out}})$. All results are based on the 5-way 1-shot miniImageNet setting.

#### 4.4.1 EFFECT OF THE NUMBER OF SUBSET SAMPLES $K$

Table 5: Varying the number of subset samples $K$. The relative time is the ratio of the time taken by the meta-learner with ConML to the time taken by the original meta-learner per epoch.

|  |  | w/o | K=1 | 2 | 4 | 8 | 16 | 32 |
|---|---|---|---|---|---|---|---|---|
| MAML w/ ConML | Acc.(%) | 48.75 | 56.25 | 56.25 | 56.08 | **57.59** | 57.40 | 57.33 |
|  | Mem.(MB) | 1331 | 2801 | 2845 | 3011 | 3383 | 4103 | 5531 |
|  | Time (relative) | 1 | 1.1 | 1.1 | 1.1 | 1.1 | 1.1 | 1.1 |
| ProtoNet w/ ConML | Acc.(%) | 48.90 | 51.03 | 51.46 | 52.04 | 52.30 | 52.34 | **52.48** |
|  | Mem.(MB) | 7955 | 14167 | 14563 | 15175 | 16757 | 19943 | 26449 |
|  | Time (relative) | 1 | 1.2 | 1.2 | 1.2 | 1.2 | 1.2 | 1.2 |

Table 5 presents the results from varying the number of subset samples $K$. Starting from $K = 1$, we observe moderate performance growth as $K$ increases, while memory usage grows linearly with $K$. Notably, there is a significant discrepancy in both performance and memory (approximately $\sim 2\times$) between the configurations without ConML and with $K = 1$. However, $K$ has minimal impact on time efficiency, assuming sufficient memory, since the processes are independent and can be parallelized.

### 4.4.2 THE DESIGN OF CONTRASTIVE LOSS

Here, we explore various design factors of the contrastive loss.

ConML optimizes the following objective: $L = L_v + \lambda L_c$, where $L_v$ is the validation loss, $L_c$ is the constrastive loss. In the previous sections, to highlight our motivation and perform a decoupled analysis, we used a simple contrastive loss $L_c = d^{\text{in}} - d^{\text{out}}$, with the natural cosine distance $\phi(x, y) = 1 - \frac{x^\top y}{\|x\|\|y\|}$. Here, we also considered a manually bounded Euclidean distance $\phi(x, y) = \text{sigmoid}(\|x - y\|)$. Beyond the simple contrastive loss, we incorporate the InfoNCE (Oord et al., 2018) loss for an episode with a batch $b$ containing $B$ tasks. The contrastive loss is defined as $L_c = -\sum_{\tau \in b} \log \left( \frac{\exp(-D_\tau^{in})}{\exp(-D_\tau^{in}) + \sum_{\tau' \in b \setminus \tau} \exp(-D_{\tau,\tau'}^{out})} \right)$, where $D_{\tau,\tau'}^{out} = \phi(e_\tau^*, e_{\tau'}^*)$. In this case, we treat negative "distance" as "similarity." For the similarity metric in InfoNCE, we experiment with both cosine distance $\phi(x, y) = 1 - \frac{x^\top y}{\|x\|\|y\|}$ and Euclidean distance $\phi(x, y) = \|x - y\|$.

Table 6: Testing accuracy (%) with varying contrastive loss form ($L_c$), distance function ($\phi$) and contrastive weight $\lambda$.

|  | $L_c$ | $\phi$ | $\lambda = 0$ | 0.01 | 0.03 | 0.1 | 0.3 | 1 |
|---|---|---|---|---|---|---|---|---|
| MAML w/ ConML | $d^{\text{in}} - d^{\text{out}}$ | Cosine | 48.75 | 52.19 | 54.43 | 56.25 | 55.82 | 47.39 |
|  |  | sigmoid(Euc) | 48.75 | 51.64 | 54.40 | 54.06 | 53.94 | 54.21 |
|  | InfoNCE | Cosine | 48.75 | 54.66 | 55.90 | **57.24** | 56.87 | 56.95 |
|  |  | Euc | 48.75 | 53.02 | 55.08 | 55.61 | 55.89 | 55.40 |
| ProtoNet w/ ConML | $d^{\text{in}} - d^{\text{out}}$ | Cosine | 48.90 | 49.16 | 51.58 | 51.03 | 50.06 | 48.81 |
|  |  | sigmoid(Euc) | 48.90 | 50.27 | 51.45 | 52.09 | 52.80 | 52.02 |
|  | InfoNCE | Cosine | 48.90 | 50.73 | 52.20 | 52.44 | 52.86 | 52.15 |
|  |  | Euc | 48.90 | 51.54 | 52.39 | 53.42 | 53.30 | **53.81** |

Table 6 presents the results. We observe that ConML can significantly improve the performance of meta-learners across a considerable range of $\lambda$, though setting $\lambda$ too high can lead to model collapse by overshadowing the original meta-learning objective. The choice of distance function varies between algorithms, with some performing better with specific functions. Additionally, InfoNCE outperforms the naive contrastive strategy, offering greater potential and reduced sensitivity to hyperparameters. These findings suggest that we may not have yet reached the full potential of ConML, and there are several promising directions for further improvement. For instance, refining batch sampling strategies to account for task-level similarities or developing more advanced subset-sampling methods could enhance performance further.

## 5 CONCLUSION AND DISCUSSION

In this work, we propose ConML, a universal, learner-agnostic contrastive meta-learning framework that emulates the alignment and discrimination capabilities integral to human fast learning, achieved through task-level contrastive learning in the model space. ConML can be seamlessly integrated with conventional episodic meta-training, and we provide specific implementations across a wide range of meta-learning algorithms. Empirical results show that ConML consistently and significantly enhances meta-learning performance by improving the meta-learner's fast-adaptation and task-level generalization abilities. Additionally, we explore in-context learning by reformulating it within the meta-learning paradigm, demonstrating how ConML can be effectively integrated to boost performance. The primary contribution of ConML is offering a learner-agnostic and efficient framework built on the most general meta-learning setting and training procedure. While the current implementation of ConML is relatively simple, it lays a foundation for general contrastive meta-learning and offers numerous opportunities for further improvement, such as optimizing sampling strategies or refining the contrastive loss function.

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

## A  SPECIFICATIONS OF META-LEARNING WITH CONML

Here, we provide the specific algorithm process of representative implementation ConML, including the universal framework of ConML (Algorithm 3), the most efficient implementation of ConMLwith $K = 1$ and $\pi_\kappa(\mathcal{D}_\tau^{\text{tr}} \cup \mathcal{D}_\tau^{\text{val}}) = \mathcal{D}_\tau^{\text{tr}}$ (Algorithm 4), training ICL model with ConML (Algorithm 5), MAML w/ ConML (Algorithm 6), Reptile w/ ConML (Algorithm 7), SCNAPs w/ ConML (Algorithm 8), ProtoNet w/ ConML (Algorithm 9).

---

**Algorithm 3** ConML.

---

**Input:** Task distribution $p(\tau)$, batch size $B$, inner-task sample times $K$ and sampling strategy $\pi_\kappa$.
**while** Not converged **do**
    Sample a batch of tasks $\boldsymbol{b} \sim p^B(\tau)$.
    **for** All $\tau \in \boldsymbol{b}$ **do**
        **for** $k = 1, 2, \cdots, K$ **do**
            Sample $\kappa_k$ from $\pi_\kappa(\mathcal{D}_\tau^{\text{tr}} \cup \mathcal{D}_\tau^{\text{val}})$;
            Get model representation $\boldsymbol{e}_\tau^{\kappa_k} = \psi(g(\kappa_k; \theta))$;
        **end for**
        Get model representation $\boldsymbol{e}_\tau^* = \psi(g(\mathcal{D}_\tau^{\text{tr}} \cup \mathcal{D}_\tau^{\text{val}}; \theta))$;
        Get inner-task distance $d_\tau^{\text{in}}$ by equation 1;
        Get task-specific model $h_\tau = g(\mathcal{D}_\tau^{\text{tr}}; \theta)$;
        Get validation loss $\mathcal{L}(\mathcal{D}_\tau^{\text{val}}; h_\tau)$;
    **end for**
    Get $d^{\text{in}} = \frac{1}{B} \sum_{\tau \in \boldsymbol{b}} d_\tau^{\text{in}}$ and $d^{\text{out}}$ by equation 2;
    Get loss $L$ by equation 3;
    Update $\theta$ by $\theta \leftarrow \theta - \nabla_\theta L$.
**end while**

---

**Algorithm 4** ConML ($K = 1$).

---

**Input:** Task distribution $p(\tau)$, batch size $B$ (inner-task sample times $K = 1$ and sampling strategy $\pi_\kappa(\mathcal{D}_\tau^{\text{tr}} \cup \mathcal{D}_\tau^{\text{val}}) = \mathcal{D}_\tau^{\text{tr}}$).
**while** Not converged **do**
    Sample a batch of tasks $\boldsymbol{b} \sim p^B(\tau)$.
    **for** All $\tau \in \boldsymbol{b}$ **do**
        Get task-specific model $h_\tau = g(\mathcal{D}_\tau^{\text{tr}}; \theta)$, and model representation $\boldsymbol{e}_\tau^{\kappa_k} = \psi(g(\kappa_k; \theta))$;
        Get model representation $\boldsymbol{e}_\tau^* = \psi(g(\mathcal{D}_\tau^{\text{tr}} \cup \mathcal{D}_\tau^{\text{val}}; \theta))$;
        Get inner-task distance $d_\tau^{\text{in}}$ by equation 1;
        Get validation loss $\mathcal{L}(\mathcal{D}_\tau^{\text{val}}; h_\tau)$;
    **end for**
    Get $d^{\text{in}} = \frac{1}{B} \sum_{\tau \in \boldsymbol{b}} d_\tau^{\text{in}}$ and $d^{\text{out}}$ by equation 2;
    Get loss $L$ by equation 3;
    Update $\theta$ by $\theta \leftarrow \theta - \nabla_\theta L$.
**end while**

---

---

**Algorithm 5** In-Context Learning with ConML (ICL w/ ConML).

---

**Input:** Task distribution $p(\tau)$, batch size $B$, inner-task sample times $K$ and sampling strategy $\pi_\kappa$, dummy input $u$ (probe).
**while** Not converged **do**
   Sample a batch of tasks $\boldsymbol{b} \sim p^B(\tau)$.
   **for** All $\tau \in \boldsymbol{b}$ **do**
     **for** $k = 1, 2, \cdots, K$ **do**
       Sample $\kappa_k$ from $\pi_\kappa(\mathcal{D}_\tau)$;
       Get $\boldsymbol{e}_\tau^{\kappa_k} = g([\vec{\kappa_k}, u]; \theta)$;
     **end for**
     Get $\boldsymbol{e}_\tau^* = g([\vec{\mathcal{D}}_\tau, u]; \theta)$;
     Get inner-task distance $d_\tau^{\text{in}}$ by equation 1;
     Get task loss $\frac{1}{m} \sum_{i=0}^{m-1} \ell(y_{\tau,i+1}, g([\vec{\mathcal{D}}_{\tau,0:i}, x_{\tau,i+1}]; \theta))$;
   **end for**
   Get $d^{\text{in}} = \frac{1}{B} \sum_{\tau \in \boldsymbol{b}} d_\tau^{\text{in}}$ and $d^{\text{out}}$ by equation 2;
   Get loss $L = \frac{1}{B} \sum_{\tau \in \boldsymbol{b}} \frac{1}{m} \sum_{i=0}^{m-1} \ell(y_{\tau,i+1}, g([\vec{\mathcal{D}}_{\tau,0:i}, x_{\tau,i+1}]; \theta)) + \lambda(d^{\text{in}} - d^{\text{out}})$;
   Update $\theta$ by $\theta \leftarrow \theta - \nabla_\theta L$.
**end while**

---

**Algorithm 6** MAML w/ ConML.

---

**Input:** Task distribution $p(\tau)$, batch size $B$, inner-task sample times $K = 1$ and sampling strategy $\pi_\kappa$
**while** Not converged **do**
   Sample a batch of tasks $\boldsymbol{b} \sim p^B(\tau)$.
   **for** All $\tau \in \boldsymbol{b}$ **do**
     **for** $k = 1, 2, \cdots, K$ **do**
       Sample $\kappa_k$ from $\pi_\kappa(\mathcal{D}_\tau^{\text{tr}} \cup \mathcal{D}_\tau^{\text{val}})$;
       Get model representation $\boldsymbol{e}_\tau^{\kappa_k} = \theta - \nabla_\theta \mathcal{L}(\kappa_k; h_\theta)$;
     **end for**
     Get model representation $\boldsymbol{e}_\tau^* = \theta - \nabla_\theta \mathcal{L}(\mathcal{D}_\tau^{\text{tr}} \cup \mathcal{D}_\tau^{\text{val}}; h_\theta)$.
     Get inner-task distance $d_\tau^{\text{in}}$ by equation 1;
     Get task-specific model $h_{\theta - \nabla_\theta \mathcal{L}(\mathcal{D}_\tau^{\text{tr}}; \theta)}$;
     Get validation loss $\mathcal{L}(\mathcal{D}_\tau^{\text{val}}; h_{\theta - \nabla_\theta \mathcal{L}(\mathcal{D}_\tau^{\text{tr}}; h_\theta)})$;
   **end for**
   Get $d^{\text{in}} = \frac{1}{B} \sum_{\tau \in \boldsymbol{b}} d_\tau^{\text{in}}$ and $d^{\text{out}}$ by equation 2;
   Get loss $L$ by equation 3;
   Update $\theta$ by $\theta \leftarrow \theta - \nabla_\theta L$.
**end while**

---

**Algorithm 7** Reptile w/ ConML.

---

**Input:** Task distribution $p(\tau)$, batch size $B$. (inner-task sample times $K = 1$ and sampling strategy $\pi_\kappa(\mathcal{D}_\tau^{\text{tr}} \cup \mathcal{D}_\tau^{\text{val}}) = \mathcal{D}_\tau^{\text{tr}})$
**while** Not converged **do**
   Sample a batch of tasks $\boldsymbol{b} \sim p^B(\tau)$.
   **for** All $\tau \in \boldsymbol{b}$ **do**
     **for** $k = 1, 2, \cdots, K$ **do**
       Sample $\kappa_k$ from $\pi_\kappa(\mathcal{D}_\tau)$;
       Get model representation $\boldsymbol{e}_\tau^{\kappa_k} = \theta - \nabla_\theta \mathcal{L}(\kappa_k; h_\theta)$;
     **end for**
     Get model representation $\boldsymbol{e}_\tau^* = \theta - \nabla_\theta \mathcal{L}(\mathcal{D}_\tau^{\text{tr}} \cup \mathcal{D}_\tau^{\text{val}}; h_\theta)$.
     Get inner-task distance $d_\tau^{\text{in}}$ by equation 1;
   **end for**
   Get $d^{\text{in}} = \frac{1}{B} \sum_{\tau \in \boldsymbol{b}} d_\tau^{\text{in}}$ and $d^{\text{out}}$ by equation 2;
   Get loss $L$ by equation 3;
   Update $\theta$ by $\theta \leftarrow \theta + \frac{1}{B} \sum_{\tau \in \boldsymbol{b}} (\boldsymbol{e}_\tau^* - \theta) - \lambda \nabla_\theta (d^{\text{in}} - d^{\text{out}})$.
**end while**

---

---

**Algorithm 8** SCNAPs w/ ConML.

---

**Note:** Here $h_w$ corresponds to the feature extractor $f_\theta$; $H_\theta$ corresponds to the task encoder $g_\phi$ in (Bateni et al., 2020).
**Input:** Task distribution $p(\tau)$, batch size $B$, inner-task sample times $K$ and sampling strategy $\pi_\kappa$.
Pretrain $h_w$ with the mixture of all meta-training data;
**while** Not converged **do**
    Sample a batch of tasks $\boldsymbol{b} \sim p^B(\tau)$.
    **for** All $\tau \in \boldsymbol{b}$ **do**
        **for** $k = 1, 2, \cdots, K$ **do**
            Sample $\kappa_k$ from $\pi_\kappa(\mathcal{D}_\tau^{\text{tr}} \cup \mathcal{D}_\tau^{\text{val}})$;
            Get model representation $\boldsymbol{e}_\tau^{\kappa_k} = H_\theta(\kappa_k)$;
        **end for**
        Get model representation $\boldsymbol{e}_\tau^* = H_\theta(\mathcal{D}_\tau^{\text{tr}} \cup \mathcal{D}_\tau^{\text{val}})$;
        Get inner-task distance $d_\tau^{\text{in}}$ by equation 1;
        Get task-specific model by FiLM $h_\tau = h_{w, H_\theta(\mathcal{D}_\tau^{\text{tr}})}$;
        Get validation loss $\mathcal{L}(\mathcal{D}_\tau^{\text{val}}; h_\tau)$;
    **end for**
    Get $d^{\text{in}} = \frac{1}{B} \sum_{\tau \in \boldsymbol{b}} d_\tau^{\text{in}}$ and $d^{\text{out}}$ by equation 2;
    Get loss $L$ by equation 3;
    Update $\theta$ by $\theta \leftarrow \theta - \nabla_\theta L$.
**end while**

---

**Algorithm 9** ProtoNet w/ ConML ($N$-way classification).

---

**Input:** Task distribution $p(\tau)$, batch size $B$, inner-task sample times $K = 1$ and sampling strategy $\pi_\kappa$
**while** Not converged **do**
    Sample a batch of tasks $\boldsymbol{b} \sim p^B(\tau)$.
    **for** All $\tau \in \boldsymbol{b}$ **do**
        **for** $k = 1, 2, \cdots, K$ **do**
            Sample $\kappa_k$ from $\pi_\kappa(\mathcal{D}_\tau^{\text{tr}} \cup \mathcal{D}_\tau^{\text{val}})$;
            Calculate prototypes $\boldsymbol{c}_j = \frac{1}{|\kappa_{k,j}|} \sum_{(x_i, y_i) \in \kappa_{k,j}} f_\theta(x_i)$ for $j = 1, \cdots, N$;
            Get model representation $\boldsymbol{e}_\tau^{\kappa_k} = [\boldsymbol{c}_1 | \boldsymbol{c}_2 | \cdots | \boldsymbol{c}_N]$;
        **end for**
        Calculate prototypes $\boldsymbol{c}_j = \frac{1}{|\mathcal{D}_j|} \sum_{(x_i, y_i) \in \mathcal{D}_j} f_\theta(x_i)$ for $j = 1, \cdots, N$;
        Get model representation $\boldsymbol{e}_\tau^* = [\boldsymbol{c}_1 | \boldsymbol{c}_2 | \cdots | \boldsymbol{c}_N]$;
        Get inner-task distance $d_\tau^{\text{in}}$ by equation 1;
        Get task-specific model $h_{[\boldsymbol{c}_1 | \boldsymbol{c}_2 | \cdots | \boldsymbol{c}_N]}$, which gives prediction by $p(y = j \mid x) = \frac{exp(-d(f_\theta(x), \boldsymbol{c}_j))}{\sum_{j'} exp(-d(f_\theta(x), \boldsymbol{c}_{j'}))}$;
        Get validation loss $\mathcal{L}(\mathcal{D}_\tau^{\text{val}}; h_{[\boldsymbol{c}_1 | \boldsymbol{c}_2 | \cdots | \boldsymbol{c}_N]})$;
    **end for**
    Get $d^{\text{in}} = \frac{1}{B} \sum_{\tau \in \boldsymbol{b}} d_\tau^{\text{in}}$ and $d^{\text{out}}$ by equation 2;
    Get loss $L$ by equation 3;
    Update $\theta$ by $\theta \leftarrow \theta - \nabla_\theta L$.
**end while**

---

# B IN-CONTEXT LEARNING WITH CONML

## B.1 IN-CONTEXT LEARNING

In-context learning (ICL) is first proposed for large language models (Brown et al., 2020), where examples in a task are integrated into the prompt (input-output pairs) and given a new query input, the language model can generate the corresponding output. This approach allows pre-trained model to address new tasks without fine-tuning the model. For example, given "*happy->positive; sad->negative; blue->*", the model can output "*negative*", while given "*green->cool; yellow->warm; blue->*" the model can output "*cool*". ICL has the ability to learn from the prompt. Training ICL can be viewed as learning to learn, i.e., meta-learning (Min et al., 2022; Garg et al., 2022; Kirsch et al., 2022). More generally, the input and output are not necessarily to be natural language. In ICL, a sequence model $T_\theta$ (typically transformer (Vaswani et al., 2017)) is trained to map sequence $[x_1, y_1, x_2, y_2, \cdots, x_{m-1}, y_{m-1}, x_m]$ (prompt prefix) to prediction $y_m$. Given distribution $P$ of training prompt $t$, then training ICL follows an auto-regressive manner:

$$\min_\theta \mathbb{E}_{t \sim P(t)} \frac{1}{m} \sum\nolimits_{i=0}^{m-1} \ell(y_{t,i+1}, T_\theta([x_{t,1}, y_{t,1}, \cdots, x_{t,i+1}])). \tag{4}$$

It has been mentioned that the training of ICL can be viewed as an instance of meta-learning (Garg et al., 2022; Akyürek et al., 2022) as $T_\theta$ learns to learn from prompt. In this section we first formally reformulate $T_\theta$ to meta-learner $g(; \theta)$, then introduce how ConML can be integrated with ICL.

## B.2 A META-LEARNING REFORMULATION

Denote a sequentialized $\mathcal{D}$ as $\vec{\mathcal{D}}$ where the sequentializer is default to bridge $p(\tau)$ and $P(t)$. Then the prompt $[x_{\tau,1}, y_{\tau,1}, \cdots, x_{\tau,m}, y_{\tau,m}]$ can be viewed as $\vec{\mathcal{D}_\tau^{tr}}$ which is providing task-specific information. Note that ICL does not specify an explicit output model $h(x) = g(\mathcal{D}; \theta)(x)$; instead, this procedure exists only implicitly through the feeding-forward of the sequence model, i.e., task-specific prediction is given by $g([\vec{\mathcal{D}}, x]; \theta)$. Thus we can reformulate the training of ICL equation 4 as:

$$\min_\theta \mathbb{E}_{\tau \sim p(\tau)} \frac{1}{m} \sum\nolimits_{i=0}^{m-1} \ell(y_{\tau,i+1}, g([\vec{\mathcal{D}}_{\tau,0:i}, x_{\tau,i+1}]; \theta)). \tag{5}$$

The loss in equation 5 can be evaluated through episodic meta-training, where each task in each episode is sampled multiple times to form $\mathcal{D}_\tau^{val}$ and $\mathcal{D}_\tau^{tr}$ to evaluate the episodic loss $L_v$ in an auto-regressive manner. The training of ICL thus follows the episodic meta-training (Algorithm 1), where the validation loss with determined $\mathcal{D}_\tau^{tr}$ and $\mathcal{D}_\tau^{val}$: $\mathcal{L}(\mathcal{D}_\tau^{val}; g(\mathcal{D}_\tau^{tr}; \theta))$, is replaced by loss validated in the auto-regressive manner: $\frac{1}{m} \sum_{i=0}^{m-1} \ell(y_{\tau,i+1}, g([\vec{\mathcal{D}}_{\tau,0:i}, x_{\tau,i+1}]; \theta))$.

## B.3 INTEGRATING CONML WITH ICL

Since the training of ICL could be reformulated as episodic meta-training, the three steps to measure ConML proposed in Section 3 can be also adopted for ICL, but the first step to obtain model representation $\psi(g(\mathcal{D}, \theta))$ needs modification. Due to the absence of an inner learning procedure for a predictive model for prediction $h(x) = g(\mathcal{D}; \theta)(x)$, representation by explicit model weights of $h$ is not feasible for ICL.

To represent what $g$ learns from $\mathcal{D}$, we design to incorporate $\vec{\mathcal{D}}$ with a dummy input $u$, which functions as a probe and its corresponding output can be readout as representation:

$$\psi(g(\mathcal{D}; \theta)) = g([\vec{\mathcal{D}}, u]; \theta), \tag{6}$$

where $u$ is constrained to be in the same shape as $x$, and has consistent value in an episode. The complete algorithm of ConML for ICL is provided in Appendix A. For example, for training a ICL model on linear regression tasks we can choose $u = \mathbf{1}$, and in pre-training of LLM we can choose $u =$"*what is this task?*". From the perspective of learning to learn, ConML encourages ICL to align and discriminate like it does for conventional meta-learning, while the representations to evaluate inner- and inter- task distance are obtained by probing output rather than explicit model weights. Thus, incorporating ConML into the training process of ICL benefits the fast-adaptation and task-level generalization ability. From the perspective of supervised learning, ConML is performing unsupervised data augmentation that it introduces the dummy input and contrastive objective as additional supervision to train ICL.

Table 7: Relative minimal error (Rel. Min. Error) and spared example number to reach the same error (Shot Spare) comparing ICL w/ and w/o ConML.

| Function (max prompt len.) | LR (10 shot) | SLR (10 shot) | DT (20 shot) | NN (40 shot) |
|:---:|:---:|:---:|:---:|:---:|
| Rel. Min. Error | $0.42 \pm 0.09$ | $0.49 \pm .06$ | $0.81 \pm 0.12$ | $0.74 \pm 0.19$ |
| Shot Spare | $-4.68 \pm 0.45$ | $-3.94 \pm 0.62$ | $-4.22 \pm 1.29$ | $-11.25 \pm 2.07$ |

Following (Garg et al., 2022), we investigate ConML on ICL by learning to learn synthetic functions including linear regression (LR), sparse linear regression (SLR), decision tree (DT) and 2-layer neural network with ReLU activation (NN). We train the GPT-2 (Radford et al., 2019)-like transformer for each function with ICL and ICL w/ ConML respectively and compare the inference (meta-testing) performance. We follow the same model structure, data generation and training settings (Garg et al., 2022). We implement ICL w/ ConML with $K = 1$ and $\pi_\kappa([x_1, y_1, \cdots, x_n, y_n]) = [x_1, y_1, \cdots, x_{\lfloor \frac{n}{2} \rfloor}, y_{\lfloor \frac{n}{2} \rfloor}]$. To obtain the implicit representation equation 6, we sample $u$ from a standard normal distribution (the same with $x$'s distribution) independently in each episode. Since the output of equation 6 is a scalar, i.e., representation $e \in \mathbb{R}$, we adopt distance measure $\phi(a, b) = \sigma((a - b)^2)$, where $\sigma(\cdot)$ is sigmoid function to bound the squared error. $\lambda = 0.02$.

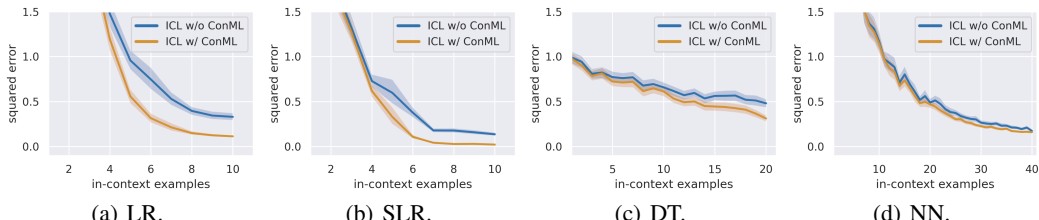

| (a) LR. | (b) SLR. | (c) DT. | (d) NN. |
|:---:|:---:|:---:|:---:|

Figure 4: In-context learning performance.

### B.4 PERFORMANCE COMPARISON

Figure 4 shows that varying the number of in-context examples during inference, ICL w/ ConML always makes more accurate predictions than ICL. Table 7 collects the two values to show the effect ConML brings to ICL: *Rel. Min. Error* is ICL w/ ConML's minimal inference error given different number of examples, divided by ICL's; *Shot Spare* is when ICL w/ ConML obtain an error no larger than ICL's minimal error, the difference between the corresponding example numbers. Note that the learning of different functions (different meta-datasets) share the same settings about ConML, which shows ConML can bring ICL universal improvement with cheap implementation. We notice that during training of LR and SLR $\lfloor \frac{n}{2} \rfloor = 5$, which happens to equals to the dimension of the regression task. This means sampling by $\pi_\kappa$ would results in the minimal sufficient information to learn the task. In this case, minimizing $d^{\text{in}}$ is particularly beneficial for the fast-adaptation ability, shown as Figure 4(a) and 4(b). This indicates that introducing prior knowledge to design the hyperparameter settings of ConML could bring more advantage. The effect of ConML on ICL is without loss of generalizability to real-world applications like pretraining large language models.

## C MORE RESULTS ON FEW-SHOT IMAGE CLASSIFICATION

Here, we provide ConML's empirical results on *mini*ImageNet, *tiered*ImageNet, and a large-scale dataset, META-DATASET (Triantafillou et al., 2020).

Table 8 shows results on META-DATASET, experimented in the same model, training (on ILSVRC-2012 only) and evaluation setting following (Triantafillou et al., 2020), and the ConML is introduced with the same setting as Section 4.2 (inner-task sampling $K = 1$ and $\pi_\kappa(\mathcal{D}_\tau^{\text{tr}} \cup \mathcal{D}_\tau^{\text{val}}) = \mathcal{D}_\tau^{\text{tr}}$, $\phi(a, b) = 1 - a \cdot b / \|a\| \|b\|$ (cosine distance) and $\lambda = 0.1$.). It can be observed that ConML also brings consistent improvement on META-DATASET.

Table 8: Experiment results on META-DATASET.

| Baseline | MatchNet | | ProtoNet | | fo-MAML | | fo-Proto-MAML | |
|---|---|---|---|---|---|---|---|---|
| ConML | w/o | w/ | w/o | w/ | w/o | w/ | w/o | w/ |
| ILSVRC | 45.0 | 51.1 | 50.5 | 52.3 | 45.5 | 54.1 | 49.5 | 54.3 |
| Omniglot | 52.2 | 54.6 | 59.9 | 61.2 | 55.5 | 63.7 | 63.3 | 69.8 |
| Aircraft | 48.9 | 51.5 | 53.1 | 54.9 | 56.2 | 64.9 | 55.9 | 61.5 |
| Birds | 62.2 | 66.8 | 68.7 | 68.9 | 63.6 | 69.9 | 68.6 | 68.6 |
| Textures | 64.1 | 67.6 | 66.5 | 68.4 | 68.0 | 72.3 | 66.4 | 69.4 |
| Quick Draw | 42.8 | 46.7 | 48.9 | 50.0 | 43.9 | 48.5 | 51.5 | 53.1 |
| Fungi | 33.9 | 36.4 | 39.7 | 40.9 | 32.1 | 40.6 | 39.9 | 43.7 |
| VGG Flower | 80.1 | 84.9 | 85.2 | 88.0 | 81.7 | 90.4 | 87.1 | 91.0 |
| Traffic Signs | 47.8 | 49.5 | 47.1 | 48.6 | 50.9 | 52.2 | 48.8 | 51.5 |
| MS COCO | 34.9 | 40.1 | 41.0 | 42.4 | 35.3 | 43.5 | 43.7 | 48.9 |

Table 9: Meta-testing accuracy on *mini*ImageNet.

| Category | Algorithm | Objective | 5-way 1-shot | 5-way 5-shot | Relative Gain | Relative Time |
|---|---|---|---|---|---|---|
| Optimization-Based | MAML | - | $48.75 \pm 1.25$ | $64.50 \pm 1.02$ | 9.16% | 1.1× |
| | | w/ ConML | $\mathbf{56.25 \pm 0.94}$ | $\mathbf{67.37 \pm 0.97}$ | | |
| | FOMAML | - | $48.12 \pm 1.40$ | $63.86 \pm 0.95$ | 12.65% | 1.2× |
| | | w/ ConML | $\mathbf{57.64 \pm 1.29}$ | $\mathbf{68.50 \pm 0.78}$ | | |
| | Reptile | - | $49.21 \pm 0.60$ | $64.31 \pm 0.97$ | 5.58% | 1.5× |
| | | w/ ConML | $\mathbf{52.82 \pm 1.06}$ | $\mathbf{67.04 \pm 0.81}$ | | |
| Metric-Based | MatchNet | - | $43.92 \pm 1.03$ | $56.26 \pm 0.90$ | 10.59% | 1.2× |
| | | w/ ConML | $\mathbf{48.75 \pm 0.88}$ | $\mathbf{62.04 \pm 0.89}$ | | |
| | ProtoNet | - | $48.90 \pm 0.84$ | $65.69 \pm 0.96$ | 3.31% | 1.2× |
| | | w/ ConML | $\mathbf{51.03 \pm 0.91}$ | $\mathbf{67.35 \pm 0.72}$ | | |
| Amortization-Based | SCNAPs | - | $53.14 \pm 0.88$ | $70.43 \pm 0.76$ | 3.12% | 1.3× |
| | | w/ ConML | $\mathbf{55.73 \pm 0.86}$ | $\mathbf{71.70 \pm 0.71}$ | | |

Table 10: Meta-testing accuracy on *tiered*ImageNet.

| Category | Algorithm | Objective | 5-way 1-shot | 5-way 5-shot | Relative Gain | Relative Time |
|---|---|---|---|---|---|---|
| Optimization-Based | MAML | - | $51.39 \pm 1.31$ | $68.25 \pm 0.98$ | 10.07% | 1.1× |
| | | w/ ConML | $\mathbf{58.75 \pm 1.45}$ | $\mathbf{72.94 \pm 0.98}$ | | |
| | FOMAML | - | $51.44 \pm 1.51$ | $68.32 \pm 0.95$ | 9.78% | 1.2× |
| | | w/ ConML | $\mathbf{58.21 \pm 1.22}$ | $\mathbf{73.26 \pm 0.78}$ | | |
| | Reptile | - | $47.88 \pm 1.62$ | $65.10 \pm 1.13$ | 10.78% | 1.5× |
| | | w/ ConML | $\mathbf{55.01 \pm 1.28}$ | $\mathbf{70.15 \pm 1.00}$ | | |
| Metric-Based | MatchNet | - | $48.74 \pm 1.06$ | $61.30 \pm 0.94$ | 11.00% | 1.2× |
| | | w/ ConML | $\mathbf{53.29 \pm 1.05}$ | $\mathbf{67.86 \pm 0.77}$ | | |
| | ProtoNet | - | $52.50 \pm 0.96$ | $71.03 \pm 0.74$ | 3.94% | 1.2× |
| | | w/ ConML | $\mathbf{54.62 \pm 0.79}$ | $\mathbf{73.78 \pm 0.75}$ | | |
| Amortization-Based | SCNAPs | - | $62.88 \pm 1.04$ | $79.82 \pm 0.87$ | 2.91% | 1.3× |
| | | w/ ConML | $\mathbf{65.06 \pm 0.95}$ | $\mathbf{81.79 \pm 0.80}$ | | |

