# OpenReview forum: "ConML: A Universal Meta-Learning Framework with Task-Level Contrastive Learning"
_ICLR.cc/2025/Conference — Submitted to ICLR 2025_

### Official Review · Reviewer_t7Bt · 2024-10-20

**Soundness:** 2
**Presentation:** 3
**Contribution:** 3
**Rating:** 5
**Confidence:** 4

**Summary:**

This paper introduces ConML, a universal meta-learning framework that incorporates task-level contrastive learning in the model space. The authors provide extensive experiments demonstrating ConML's effectiveness across different meta-learning approaches and tasks. While the paper is well-written and the empirical evaluation is thorough, there are some concerns about the novelty and positioning of the contribution.

**Strengths:**

1. The paper is well-written and clearly presented.
2. The experimental evaluation is extensive, covering multiple datasets and meta-learning approaches.

**Weaknesses:**

1. The relationship between the proposed model-space contrastiveness and existing feature-space contrastiveness in meta-learning is not fully explored or differentiated. (see comment below)
2. Some key comparisons to related contrastive meta-learning approaches are missing. (see comment below)

**Questions:**

1. The proposed algorithm seems to be a generalized version of feature-contrastive learning. Previous works like [1] and [2] have shown a close relationship between meta-learning and contrastive learning. Optimization-based models implicitly contain feature information [2], and metric-based prototypes directly encode feature information. In this sense, contrasting models may be inherently similar to contrasting features. The authors should provide more context on how model contrastiveness differs from or extends feature contrastiveness. Clarifying this relationship is important, even if model contrastiveness is a generalization.
2. The paper lacks comparisons with other contrastive-based meta-learning algorithms such as ANIL [1] and MAML with the Zeroing trick [2]. Including these baselines would help position the contribution more clearly.
3. It would be informative to see how the model alignment dynamics evolve during training. Plotting alignment metrics (e.g., d^in, d^in - d^out, d^out) for MAML and ConML could provide insights into the alignment/discrimination process.
4. An ablation study exploring different meta-contrastive objectives (e.g., only d^in or only -d^out) could help gain more insight into the relative importance of alignment vs. discrimination for meta-learning.

---

### Official Review · Reviewer_JELZ · 2024-10-24

**Soundness:** 2
**Presentation:** 3
**Contribution:** 2
**Rating:** 3
**Confidence:** 5

**Summary:**

This paper introduces a learner-agnostic framework aimed at improving the performance of meta-learning algorithms by leveraging alignment and discrimination across tasks. Specifically, it incorporates a contrastive loss during meta-training to minimize the distance across different subsets of the same task, and maximize the inter-task distance. The authors demonstrate the effectiveness of the proposed ConML approach through experiments across several types of meta-learning paradigms, including optimization-based, metric-based, amortization-based, and in-context learning methods.

**Strengths:**

*  The proposed approach can be easily integrated into various meta-learning frameworks with minimal additional computational overhead.
*  Experimental results indicate that ConML increases the performance of all the evaluated meta-learning methods.
*  Incorporating a task-level contrastive loss into meta-learning frameworks improves the learned feature representation.

**Weaknesses:**

*  The paper could benefit from including more recent references, particularly those that study the combination of meta-learning and contrastive learning [1]. Relevant previous work [2,3,4] have already explored this area and should be considered in both the experimental setup and the discussion section.
*  The notation is not consistent throughout the text. For instance, the model $h$ is referred to as $h_\theta$, $h_{\theta, D}$, and $h_t$ in the paper, where $\theta$ are the model parameters, $D$ is a dataset, and $t$ is a task
*  More training details are necessary to ensure reproducibility, particularly for the ICL experiments. It is not clear if these experiments are performed in the in-domain or cross-domain scenario.
*  The plots in Figures 2.b and 2.e could be enlarged or zoomed in on a smaller x-axis interval for better clarity.
*  The results do not include standard deviations, making it difficult to assess whether the improvements with ConML are statistically significant.
*  The contribution of the paper is somewhat limited as it primarily adapts an existing contrastive learning approach to meta-learning frameworks, an idea that has already been explored to some extent in previous literature [1,2,3,4].




References:\
[1] Ni, Renkun, et al. "The close relationship between contrastive learning and meta-learning." International conference on learning representations. 2021.\
[2] Lee, Dong Bok, et al. "Self-supervised set representation learning for unsupervised meta-learning." The Eleventh International Conference on Learning Representations. 2023.\
[3] Li, Chengyang, et al. "MetaCL: a semi-supervised meta learning architecture via contrastive learning." International Journal of Machine Learning and Cybernetics 15.2 (2024): 227-236.\
[4] Tian, Pinzhuo, and Yang Gao. "Improving meta-learning model via meta-contrastive loss." Frontiers of Computer Science 16.5 (2022): 165331.

**Questions:**

*  Why is the proposed approach described as “universal”? It seems that it must be adapted to each specific meta-learning method, such as by defining $\psi$. Moreover, it cannot be applied universally to solve any task, but it is evaluated only on in-domain scenarios considering tasks sampled from the same dataset during meta-training and meta-testing.
*  Does the projection function $\psi:H→\mathbb{R}^d$ need to satisfy certain properties? A more in depth explanation, also with some mathematical proof, and references, might be useful for the reader.
*  Given that the projection function $\psi$ is method-dependent, would it be possible to define a universal function (or neural network) that could be applied across all meta-learning frameworks? This would align better with the learner-agnostic claim.
*  In line 107, the authors provide a definition of meta-learning assuming that the meta-learned model generalizes to a new task $\tau'$ sampled from $p(\tau')$, which is different from the task distribution $p(\tau)$ used during meta-training. This is not common in "standard" meta-learning, as one of the assumption is that all tasks (both for meta-train and meta-test) are sampled from the same probability distribution. Do the authors refer here to the cross-domain scenario, instead? Is the proposed approach applicable also in this challenging scenario?
* Is there evidence or experimental support for the claim that $d_{out}$ enhances generalization? All experiments appear to be performed considering the in-domain scenario, which does not fully demonstrate generalization capabilities.
*  Do the tests performed with CAML w/ConML consider the cross-domain scenario? Specifically, was meta-training performed on the same dataset combinations as in CAML (ImageNet-1k, Fungi, MSCOCO, and WikiArt) and then evaluated on the datasets in Table 3? If not, the results in Table 3 may not be directly comparable.
*  Regarding the experiments in Figure 2.c, could you clarify why MAML’s performance remains unchanged as the number of shots increases? Additionally, why does MAML with ConML and MAML with $d^{in}$​ perform better with fewer shots than with more? I expected the performance to improve with more shots at test time. Also, I recommend conducting experiments with the same number of shots during meta-training and meta-testing.

---

### Official Review · Reviewer_HQfu · 2024-10-30

**Soundness:** 2
**Presentation:** 3
**Contribution:** 2
**Rating:** 3
**Confidence:** 5

**Summary:**

In this paper, the authors propose a task-level contrastive learning framework to enhance existing meta-learning methods. To achieve this goal, the authors need to construct positive and negative samples in the model space, which is discussed in this paper.

**Strengths:**

Meta-learning aims to acquire cross-task meta-knowledge, enabling models to adapt quickly and generalize effectively to new tasks. I believe that utilizing task-level information is crucial for further enhancing existing meta-learning models. The authors explore this issue and achieve notable results in improving various meta-learning methods

**Weaknesses:**

However, defining and describing task-level information is much more challenging than at the sample level. Although the paper introduces some methods, it does not provide a common and reasonable approach for representing models in model space. For instance, in MAML, it uses the influence of the task-specific model on the meta-parameters, while in ProtoNet, it utilizes sample-level information as task-level information, which feels overly customized. Additionally, I think the use of cosine or Euclidean distance to measure model similarity to be inadequate.

**Questions:**

1. In model space, as in MAML, the weights of the task-specific model may consist of multiple weight matrices, rather than a single weight vector as in regression. How is this complexity addressed
2. Bilevel meta-learning is also an important approach that the authors do not mention
3. In my opinion, MAML and ProtoNet aim to establish a common initialization or feature space across tasks. However, in this paper, the use of d_out disrupts this objective, potentially leading to instability during training
4. In a few-shot learning setting, the task-specific learning model cannot be fully optimized. Therefore, directly using e* as a reference does not seem advisable.

---

### Official Review · Reviewer_pJ1e · 2024-11-03

**Soundness:** 2
**Presentation:** 3
**Contribution:** 2
**Rating:** 5
**Confidence:** 4

**Summary:**

This work proposes a new meta-learning framework that enhances model adaptability through task-level contrastive learning, allowing it to distinguish and align tasks effectively. Applicable across various meta-learning approaches, ConML improves task generalization by minimizing inner-task distance and maximizing inter-task distance, demonstrating substantial gains in few-shot learning tasks like regression, image classification, and molecular property prediction.

**Strengths:**

1. ConML is flexible and general, which can be seamlessly integrated into various meta-learning algorithms (optimization-based, metric-based, amortization-based, and in-context learning) without relying on specific model architectures or target models.

2. ConML integrates efficiently into episodic training, requiring only lightweight additional calculations. This design allows it to achieve performance gains with minimal impact on computational resources.

3. Applying meta-learning to in-context learning is a good attempt, and its value would be greater if meta-learning could enhance the performance of current mainstream models and tasks.

**Weaknesses:**

1. The motivation for introducing task-level contrastive learning into meta-learning is not very clear. The paper briefly states that this introduction helps to "enhance both alignment and discrimination abilities." If it could clarify which specific pain points in current meta-learning this contrastive learning approach addresses and which persistent challenges in meta-learning it successfully overcomes, the value of this work would be significantly greater.

2. The performance of ConML depends on several hyperparameters (e.g., the weight parameter $\lambda$, sampling strategies, and distance functions). Manual tuning of these settings for different tasks could make the framework less user-friendly and time-intensive to optimize for best results.

3. ConML has been tested primarily on small to medium-scale models. Its effectiveness on larger architectures, like deep convolutional networks or transformers, remains unproven, potentially limiting its scalability in more computationally demanding applications.

4. ConML has been validated mostly on few-shot learning tasks like classification and regression, with limited testing on more complex tasks such as object detection, segmentation, or NLP tasks. This focus restricts the scope of its demonstrated effectiveness across broader real-world applications.

**Questions:**

1. Which specific pain points in current meta-learning does task-level contrastive learning address and which challenges in meta-learning can it overcome?

2. What's an efficient way to tune certain hyperparameters (e.g., the weight parameter $\lambda$, sampling strategies, and distance functions)?

3. How effective is ConML on larger architectures, such as deep convolutional networks or transformers?

4. How would this framework perform in experiments on more realistic tasks, like segmentation or detection?

Please see Weaknesses for details.

---

### Meta-Review · Area_Chair_tHX1 · 2024-12-15

**Metareview:**

While ConML introduces a promising and innovative concept of task-level contrastive learning, the paper has several weaknesses in its motivation, empirical validation, and comparisons with related work. With the current state of the paper, a reject is suggested. However, addressing the outlined concerns could make this work a strong contribution to the meta-learning community.

**Additional Comments On Reviewer Discussion:**

No rebuttal was found.

---

### Decision · Program_Chairs · 2025-01-22

Reject